

# Identification of hub genes and prediction of the ceRNA network in adult sepsis

Kangyi Xue[1,*], Kan Wu[2,*], Haoxian Luo[3], Haihua Luo[2], Zhaoqian Zhong[2], Fen Li[4], Lei Li[5] and Li Chen[2,3]

[1] Department of Urology, The Third Affiliated Hospital of Southern Medical University, Southern Medical University, Guangzhou, Guangdong Province, China

[2] Guangdong Provincial Key Laboratory of Proteomics, State Key Laboratory of Organ Failure Research, Department of Pathophysiology, School of Basic Medical Sciences, Southern Medical University, Guangzhou, Guangdong Province, China

[3] Department of Anesthesiology, The Third Affiliated Hospital of Southern Medical University, Southern Medical University, Guangzhou, Guangdong Province, China

[4] Department of ICU, The Third Affiliated Hospital of Southern Medical University, Southern Medical University, Guangzhou, Guangdong Province, China

[5] Institute of Infection and Immunity, Henan Academy of Innovations in Medical Science, Institute of Infection and Immunity, Zhengzhou, Henan, China

* These authors contributed equally to this work.

Corresponding authors
Lei Li, lilei747667911@i.smu.edu.cn
Li Chen, chenli4618@163.com

## ABSTRACT

**Background**. Sepsis refers to a dysregulated host immune response to infection. It carries a high risk of morbidity and mortality, and its pathogenesis has yet to be fully elucidated. The main aim of this study was to identify prognostic hub genes for sepsis and to predict a competitive endogenous RNA (ceRNA) network that regulates the hub genes.

**Methods**. Six transcriptome datasets from the peripheral blood of septic patients were retrieved from the Gene Expression Omnibus (GEO) database. The robust rank aggregation (RRA) method was used to screen differentially expressed genes (DEGs) across these datasets. A comprehensive bioinformatics investigation was conducted, encompassing Gene Ontology (GO) and Kyoto Encyclopedia of Genes and Genomes (KEGG) pathway enrichment analyses using the "clusterProfiler" package in R, as well as gene set enrichment analysis (GSEA) to further elucidate the biological functions and pathways associated with the DEGs. Weighted gene co-expression network analysis (WGCNA) was performed to identify a module significantly associated with sepsis. Integration of this module with protein–protein interaction (PPI) network analysis facilitated the identification of five hub genes. These hub genes were subsequently validated using an independent dataset and reverse transcription-quantitative polymerase chain reaction (RT-qPCR) analysis of peripheral blood samples from septic patients. The prognostic values of these hub genes were assessed *via* receiver operating characteristic (ROC) curve analysis. Finally, a ceRNA network regulating the prognostic hub genes was constructed by integrating data from a literature review as well as five online databases.

**Results**. RRA analysis identified 164 DEGs across six training cohorts. Bioinformatics analyses revealed concurrent hyperinflammation and immunosuppression in sepsis patients. Five hub genes were identified *via* WGCNA and PPI network analysis, and their differential expression was verified by the validation dataset (GSE28750) and RT-qPCR analysis in the peripheral blood of septic patients. ROC analysis confirmed four

hub genes with prognostic value, and a ceRNA network was predicted to elucidate their regulatory mechanisms.

**Conclusion**. This study identified four hub genes (CLEC4D, GPR84, S100A12, and HK3) with significant prognostic value in sepsis and predicted a ceRNA network (NEAT1-hsa-miR-495-3p-ELF1) regulating their expression. The integrated analysis reconfirmed the concurrent presence of hyperinflammation and immunosuppression in hospitalized sepsis patients. These findings enhance the understanding of sepsis pathogenesis and identify potential therapeutic targets.

## INTRODUCTION

According to the Third International Consensus Definitions for Sepsis and Septic Shock (Sepsis-3), sepsis is defined as a life-threatening condition of organ dysfunction caused by a dysregulated host immune response to an infection (*Singer et al., 2016*). Currently, sepsis remains a major cause of morbidity and mortality worldwide (*Arora, Mendelson & Fox-Robichaud, 2023*). Approximately 48.9 million sepsis cases and 11 million sepsis-related mortalities were recorded worldwide in 2017, accounting for 19.7% of all global mortalities that year (*Markwart et al., 2020*). The global burden of sepsis is projected to rise in the coming decades, driven by several key factors, including aging populations with weakened immune defenses, the increasing complexity of medical interventions (*e.g.*, invasive surgeries, immunosuppressive therapies), and the compounding effects of global warming (*Wiersinga & Van der Poll, 2022*). Furthermore, sepsis survivors are at an increased risk of death or a reduced health-related quality of life even after discharge from the hospital (*Yende et al., 2016*).

The pathogenesis of sepsis is complex and dynamic, with immune dysfunction playing a key role (*Huang, Cai & Su, 2019*). Dysregulated host immune response is characterized by an initial hyperinflammatory phase followed by a sustained immunosuppressive phase, which ultimately leads to multi-organ dysfunction and mortality (*Martin, Badovinac & Griffith, 2020*; *Van der Poll et al., 2017*). Certain anti-inflammatory (anti-TNF-α, recombinant-activated protein C, and TLR4/MD2 antagonist) and immunostimulatory (immunoglobulin) agents are currently being tested in clinical trials for treating various stages of sepsis and septic shock (*Opal et al., 2013*; *Rello et al., 2017*). However, no satisfactory therapeutic outcomes have been obtained thus far, which suggests that the pathophysiology of sepsis has not yet been fully elucidated. Therefore, further research on the etiology of sepsis is needed to develop more effective treatments.

Considering that the mechanisms of sepsis are complex and that there are currently no particularly effective drugs, early recognition of sepsis can improve the survival of septic patients. The World Health Organization indicated that a significant proportion of sepsis-related mortalities could be prevented by early treatment (*Teggert, Datta & Ali,*

**Table 1  Microarray information.**

| GEO ID | Platform | Participants | Tissues | Attribute |
|---|---|---|---|---|
| GSE137340 | GPL10558 | 23 cases and 12 controls | Peripheral Blood | Training set |
| GSE69063 | GPL19983 | 19 cases and 11 controls | Peripheral Blood | Training set |
| GSE69528 | GPL10558 | 83 cases and 28 controls | Peripheral Blood | Training set |
| GSE54514 | GPL6947 | 28 cases and 18 controls | Peripheral Blood | Training set |
| GSE57065 | GPL570 | 28 cases and 25 controls | Peripheral Blood | Training set |
| GSE95233 | GPL570 | 51 cases and 22 controls | Peripheral Blood | Training set |
| GSE28570 | GPL10558 | 10 cases and 20 controls | Peripheral Blood | Validation set |

*2020*). A previous study demonstrated that rapid completion of a 3-hour bundle of sepsis care and rapid administration of antibiotics were associated with lower risk-adjusted in-hospital mortality (*Seymour et al., 2017*). However, existing research indicates that early diagnosis has been a significant challenge due to multiple comorbidities and the absence of an effective prediction technique (*Zhai et al., 2020*). Therefore, there is an urgent need to identify novel biomarkers associated with sepsis to facilitate early diagnosis, monitoring, and therapeutic intervention.

High-throughput sequencing methods have provided unprecedented insights into the study of disease mechanisms and biomarker identification (*Tabone et al., 2018*). Transcriptomics combined with bioinformatics can discover potential molecules and pathways from macroscopic aspects, guiding subsequent experiments (*Xu et al., 2022*). To screen hub genes with diagnostic and prognostic significance and to further understand the molecular mechanisms of sepsis, this study integrated bioinformatics analyses on training datasets, followed by validation using datasets and peripheral blood samples from patients. Four hub genes with prognostic significance were identified, and a competing endogenous RNA (ceRNA) network regulating the hub genes was predicted. The present study contributes to the comprehensive understanding of the molecular mechanisms underlying sepsis and may provide novel therapeutic targets.

## MATERIALS AND METHODS

### Datasets and sample selection

Gene expression datasets related to sepsis were searched *via* the Gene Expression Omnibus (GEO) database (https://www.ncbi.nlm.nih.gov/geo/; *Barrett et al., 2013*). The present study systematically retrieved microarray datasets based on the following terms: 'Sepsis', 'Adult', 'Homo sapiens', and 'Microarray'. Datasets were acquired based on the following eligibility criteria: (1) containing ≥5 sepsis patients and ≥5 healthy controls; and (2) containing raw data or gene expression by array accessible through GEO. Seven GEO datasets (GSE137340, GSE69063, GSE69528, GSE54514, GSE57065, GSE95233, and GSE28750) were included (Table 1). The series matrix file(s) or raw data were downloaded from the GEO website, along with the corresponding annotation documents from GEO or Bioconductor (https://bioconductor.org/).

## Robust rank aggregation analysis

Statistically changed genes and differentially expressed genes (DEGs) were screened using the robust rank aggregation (RRA) method, which is an unbiased and effective approach for integrating data from diverse chip platforms using a comprehensive ranking list algorithm (*Jia & Zhai, 2019*; *Yan et al., 2018*). After identifying up-ranked and down-ranked gene lists of each dataset based on the fold-change in expression levels between patients with sepsis and healthy controls, the R 'Robust Rank Aggregation' package was used to integrate all of the ranked gene lists. The adjusted *P*-value in the analysis informed the ranking of each gene in the final list. Finally, the 'limma' (v.3.54.2) and 'RobustRankAggreg' (v.1.2.1) R packages were used to screen statistically changed genes (adjusted (adj.) *P* < 0.05) and DEGs (adj. *P* < 0.05 and |log fold-change| > 1.2).

## Function and pathway enrichment analyses

To explore the impact of these genes on the development of sepsis, gene set enrichment analysis (GSEA), Gene Ontology (GO) analysis, and Kyoto Encyclopedia of Genes and Genomes (KEGG) pathway enrichment analysis were performed *via* the 'clusterProfiler' (v. 4.6.2) R package (*Wu et al., 2021*). The list 'c2.cp.reactome.v2022.1.Hs.symbols.gmt' from the Molecular Signatures Database (https://www.gsea-msigdb.org/gsea/msigdb/) was used as a reference dataset (*Liberzon et al., 2015*). Ridgeline plots and normalized enrichment scores (NES) were used to present the results of the GSEA analysis. GO contained the biological process (BP), cellular component (CC), and molecular function (MF) categories. Adjusted *P* < 0.05 was set as the threshold criterion.

## Evaluation of immune cell infiltration in peripheral blood

Based on the gene expression matrix, cell-type identification *via* the estimation of relative subsets of RNA transcripts (CIBERSORT) was used to analyze the percentage of immune cells in peripheral blood. The expression matrix was standardized by quantile normalization, and the operational mode was configured as batch mode, concerning the known set LM22 (*Newman et al., 2015*). *P* < 0.05 was considered to indicate a statistically significant difference. The Mann–Whitney *U* test was used to evaluate significant differences in infiltrating immune cell types between individuals with sepsis and healthy controls. The 'ggpubr' (v. 0.6.0) package was used to generate boxplots showing immune cell infiltration differentiation.

## Weighted gene co-expression network analysis

Weighted gene co-expression network analysis (WGCNA) is a data exploration tool that can identify key gene modules using unsupervised clustering without a priori-defined gene set (*Liu et al., 2017*). The R package 'WGCNA' (v. 1.72.1) was used in the present study to construct a co-expression network based on all expressed genes in GSE69528. Genes were divided into different modules using the topological overlap matrix-based dissimilarity measure. Key modules were identified by setting the soft-thresholding power ($\beta$) as 7 ($R^2 = 0.85$), cut height as 0.25, minimal module size as 30, and threshold as 0.2. Genes with similar expression profiles were aggregated into the same cluster *via* a dynamic tree cut algorithm. The gene modules most strongly associated with sepsis were selected

through the computation of their association with phenotype. A gene set was obtained *via* Venn analysis of DEGs from the RRA analysis and genes in the module for subsequent evaluation.

## Protein–protein interaction network analysis

The STRING database (https://cn.string-db.org/) is an online resource for accessing established proteins and predicting protein–protein interactions (PPIs), encompassing both direct physical associations between proteins and their indirect functional connections (*Szklarczyk et al., 2021*). In the present study, a PPI network was constructed using the STRING online tool with default filter conditions (combined score > 0.4). Subsequently, Cytoscape 3.9.0 (https://cytoscape.org/) software was used to optimize the PPI network for enhanced visualization (*Majeed & Mukhtar, 2023*). Plug-in minimal common oncology data elements (MCODE; https://apps.cytoscape.org/apps/mcode) was used to screen important gene clusters (Set 1). Plug-in cytoHubba-maximal clique centrality (MCC; https://apps.cytoscape.org/apps/cytohubba) with default parameters was also used to identify key genes (Set 2) in this network (*Chin et al., 2014*). Plug-in iRegulon (v.1.3; http://apps.cytoscape.org/apps/iRegulon) was used with default parameters to identify the transcription factor (TF) of Set 2 (*Janky et al., 2014*).

## Peripheral blood samples from patients with sepsis and health controls

Patients were included in the study if they were 20 years of age or older and had been diagnosed with sepsis according to the Third International Consensus Definitions for Sepsis and Septic Shock (Sepsis-3; *Singer et al., 2016*). The present study included nine septic patients and nine healthy controls. Healthy controls composed of laboratory and hospital employees were recruited, whereas patients with sepsis were admitted to the Intensive Care Unit (ICU) of The Third Affiliated Hospital of Southern Medical University. Peripheral blood samples were collected through venipuncture. The period of recruitment for this study was from November 1 to December 15, 2023. Written informed consent was obtained from the patient or the patient's family. The research protocol and use of samples were both approved and formally authorized by the Clinical Trial Ethics Committee of The Third Affiliated Hospital of Southern Medical University (approval no. 2023073; Guangzhou, China) on October 30, 2023.

## Animal care and housing

SPF-grade C57BL/6 healthy mice were purchased from Southern Medical University. Upon arrival, the animals were acclimatized for five days prior to the start of the experiments. Mice were provided ordinary Specific Pathogen Free (SPF) food and water freely throughout the experiment. All mice were housed in units of equal size and material and were exposed to 12 h light-12 h darkness-cycles. The feeding temperature was $22(\pm 2)$ °C and the humidity was 40~70%. All experiments were conducted using male adult mice (6~8 weeks old, weighing 20~22 g) during the light cycle from 10:00 am to 5:00 pm. All surviving animals at the conclusion of the experiment were to be euthanized by rapid neck dislocation, but this was unnecessary as all mice died after blood collection.

## Animal model of cecal ligation and puncture

Animal experiments were approved by the Animal Welfare and Ethics Committee of Southern Medical University, Guangzhou, China (approval no. L2018235) and were conducted in accordance with the Guide for the Care and Use of Laboratory Animals. Sixteen SPF-grade male C57BL/6 mice were randomly divided into two groups, cecal ligation and puncture (CLP) and sham, with eight mice in each group. CLP modeling of sepsis was performed according to a previously established protocol (*Li et al., 2022*). In brief, C57BL/6 mice were anesthetized with isoflurane at 5% for induction and 2.5% for maintenance. After confirming loss of consciousness, a 2-cm midline incision was made on the anterior abdomen to expose the cecum, and it was then ligated at a site 1 cm from the end of the cecum. The cecum was twice punctured with an 18-gauge needle between the ligation site and the end of the cecum, and a tiny amount of cecal content was then extruded. Following the cecum's return to the abdomen, wound clips were used to seal the incision. With the exception of the cecum puncture and ligation, the sham group received the same care as the CLP group. During the operation, if any signs of pain were observed, appropriate care was immediately provided and the necessary amount of anesthesia was adjusted accordingly. For fluid replacement, each animal received a subcutaneous injection of 1 mL of sterile saline. Afterwards, the mice were placed on thermal blankets to facilitate their recovery from anesthesia. At twenty-four hours after surgery, all surviving mice were again anaesthetized and peripheral blood was collected from the inferior vena cava. The samples were immediately placed in an ice box for storage and were processed for the subsequent experiments within one hour.

## RNA extraction and reverse transcription-quantitative polymerase chain reaction

RNA was extracted from peripheral blood samples of patients with sepsis and healthy controls using a GeneJET Stabilized and Fresh Whole Blood RNA Kit (cat. no. K0871; Thermo Fisher Scientific, Inc., Waltham, MA, USA), and single-stranded cDNA was synthesized from 1 μg of RNA using a ReverTra Ace qPCR RT Kit (cat. no. FSQ-101; Toyobo Life Science, Osaka, Japan). The reaction condition was as follows: 37 °C for 15 min and 98 °C for 5 min. The primers of target genes were acquired from PrimerBank (https://pga.mgh.harvard.edu/primerbank/) and were synthesized at BGI Genomics (Table S1). Quantification of gene expression was conducted by reverse transcription-quantitative polymerase chain reaction (RT-qPCR) using SYBR qPCR Mix (cat. no. QPK-201; Toyobo Life Science) on a 7500 Fast Real-Time PCR System (Applied Biosystems; Thermo Fisher Scientific, Inc.). The polymerase chain reaction (PCR) cycling parameters were as follows: initial denaturation at 95 °C for 3 min followed by 40 cycles of PCR reaction at 95 °C for 10 s, 60 °C for 10 s, and 72 °C for 30 s. The reaction system was set up as follows: SYBR Green Mix (5 μL), forward primer (0.2 μL), reverse primer (0.2 μL), cDNA template (0.1 μL), and ddH2O (4.5 μL), resulting in a total volume of 10 μL. PCR amplification was performed once for each sample, and the mRNA expression of target genes was normalized to 18S rRNA. Relative expression was determined using the $2^{-\Delta\Delta CT}$ method (*Zang et al., 2023*): $\Delta\Delta CT = (CT_{Target} - CT_{18srRNA})_{Sample} - (CT_{Target} - CT_{18srRNA})_{Control}$.

Peripheral blood mononuclear cells were extracted from patients with sepsis and mice after CLP using the Peripheral Blood Mononuclear Cell Isolation Kit (cat. no. P8680; Beijing Solarbio Science & Technology Co., Ltd., Beijing, China) according to the manufacturer's instructions, and subsequent steps were executed as described above.

### Receiver operating characteristic analysis

To evaluate the clinical prognostic significance of the identified hub genes in sepsis, the GSE54514 dataset was used, which includes survivor and non-survivor data. Receiver operating characteristic (ROC) curves were plotted using the R 'pROC' (v. 1.18.4) package (*Jiang et al., 2022*). The area under the curve (AUC) was calculated for each hub gene to evaluate its performance. A set of guidelines was established to determine the accuracy of different diagnostic criteria, with each set of criteria having excellent ($0.9 \leq AUC < 1$), good ($0.8 \leq AUC < 0.9$), moderate ($0.7 \leq AUC < 0.8$), poor ($0.6 \leq AUC < 0.7$) or insufficient ($0.5 \leq AUC < 0.6$) accuracy. If $AUC > 0.8$, the hub genes were regarded to have excellent sensitivity and specificity in differentiating between sepsis survivors and non-survivors.

### Correction analysis

The 'Immunity' gene list was downloaded from the GeneCards database (https://www.genecards.org/), which is a searchable, integrative database that provides comprehensive, user-friendly information on all annotated and predicted human genes. Next, the 'corrplot' (v. 0.92) R package was used to visually represent the correlation matrix (*Pei et al., 2020*). $P < 0.05$ was considered to indicate a statistically significant difference.

### CeRNA network construction

Four online microRNA (miRNA or miR) websites, namely miRWalk 3.0 (http://mirwalk.umm.uni-heidelberg.de/), microT v5 (http://diana.imis.athena-innovation.gr/DianaTools/index.php?r=MicroT_CDS/index), miRDB (https://mirdb.org/), and TargetScan Human 8.0 (https://www.targetscan.org/vert_80/), were used to predict target miRNAs (*Agarwal et al., 2015*; *Chen & Wang, 2020*; *Dweep, Gretz & Sticht, 2014*; *Paraskevopoulou et al., 2013*). The target miRNAs were selected from the intersection of $\geq 3$ databases. Interactions between long noncoding RNAs (lncRNAs) and the selected miRNAs were predicted using StarBase v2.0 (https://rnasysu.com/encori/; *Li et al., 2014*). Cytoscape software was used to construct and visualize the interaction networks of mRNAs, miRNAs, and lncRNAs.

### Statistical analysis

R version 4.2.0 (https://www.r-project.org/) was used to conduct statistical analyses. Paired $t$-tests were used to compare differences between two groups. Data are presented as the mean $\pm$ standard error of the mean. Statistical analysis and image construction were performed using GraphPad Prism 8.0.2 (GraphPad; Dotmatics, Boston, MA, USA). Figures were edited using Adobe Illustrator (AI CC2018) software (Adobe Systems, Inc., San Jose, CA, USA). Statistical significance was detected at the 0.05 level.

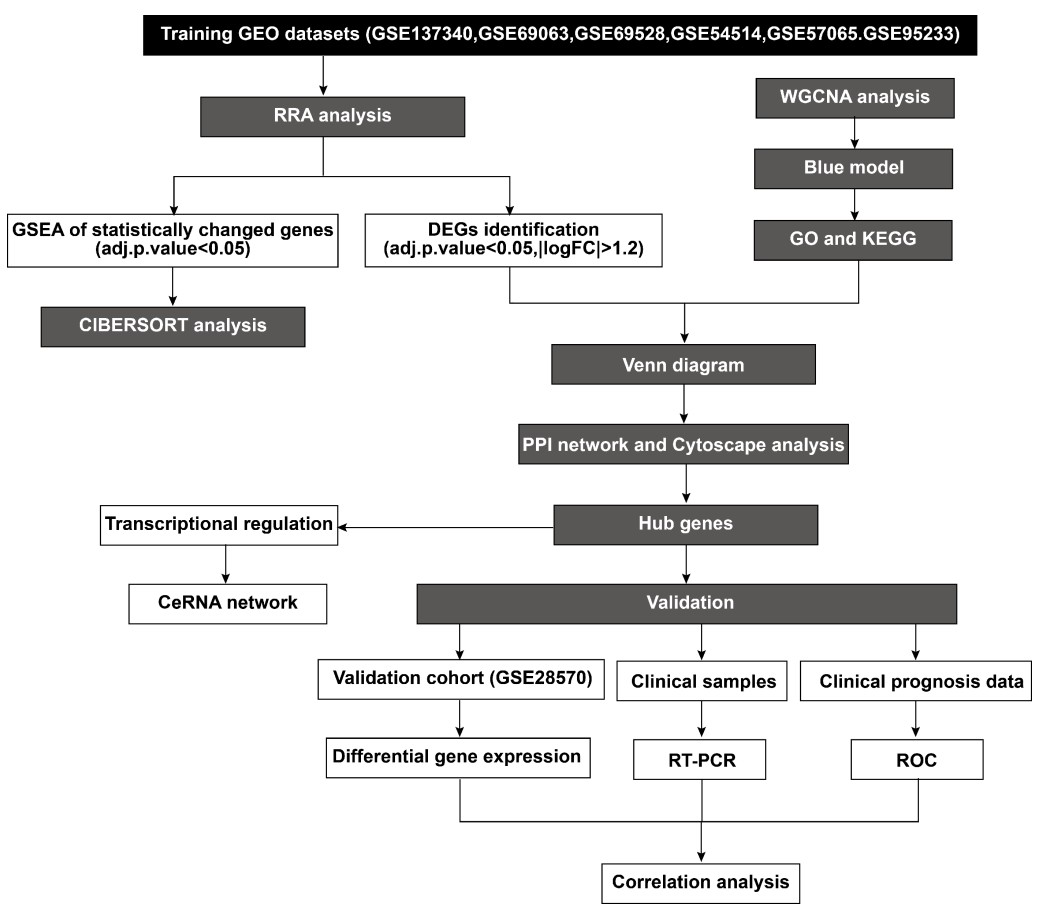

**Figure 1  Flowchart of the study.**

## RESULTS

### Information on included microarrays

According to the established inclusion criteria, the GSE137340, GSE69063, GSE69528, GSE54514, GSE57065, and GSE95233 expression datasets were used for analysis. Prior to the analysis, these datasets underwent batch effect removal followed by normalization and log2 transformation. Next, the gene expression profiles were visualized through boxplots, while significant differences between septic patients and healthy controls were shown by scatter plots in principal component analysis (Figs. S1A–1F). This study included 232 patients with sepsis and 116 healthy controls across six datasets. The basic workflow of the present study is shown in Fig. 1.

### Functional enrichment analysis of the trained differential expression gene set based on RRA

A total of 361, 673, 674, 603 590, and 912 DEGs were identified in the GSE137340, GSE69063, GSE69528, GSE54514, GSE57065, and GSE95233 datasets, respectively (Fig. 2A). Subsequent analysis of six cohorts using RRA identified 95 upregulated and

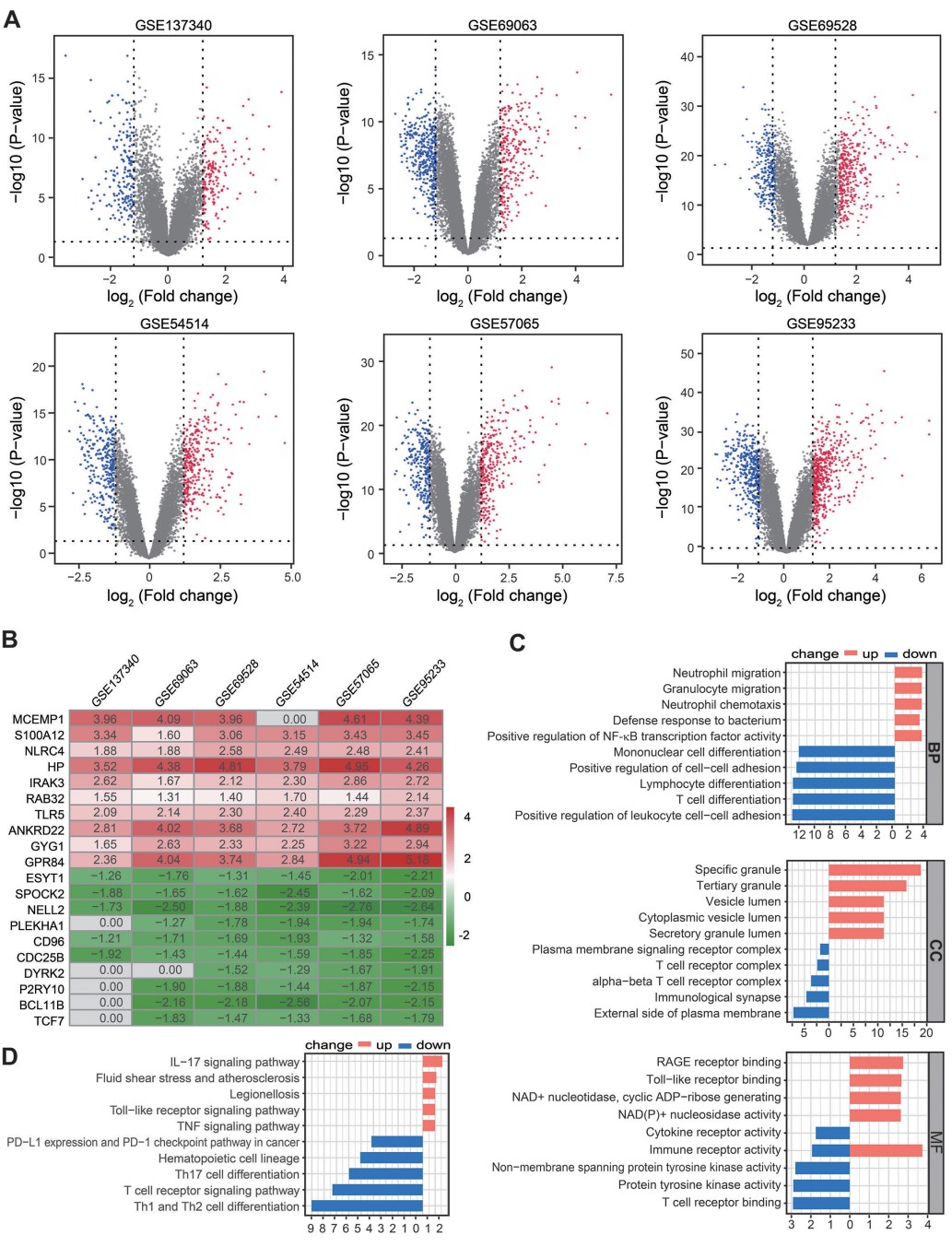

**Figure 2** **Identification of DEGs by RRA analysis.** (A) Volcano plots showing DEGs between septic patients and healthy controls across six Gene Expression Omnibus datasets. Red points indicate upregulated genes in sepsis, while blue points indicate down-regulated genes in sepsis. Gray points indicate genes with no significant difference. (B) Heatmap showing the top 10 up- and down-regulated DEGs in RRA. Red indicates highly expressed genes in sepsis, while green indicates poorly expressed genes in sepsis. (C) The enriched biological functions in Gene Ontology analysis demonstrated the underlying significance of up- and down-regulated DEGs across biological process, cellular component, and molecular function categories. (D) Top 10 pathways enriched in up- and down-regulated DEGs, as according to Kyoto Encyclopedia of Genes and Genomes. DEGs, differentially expressed genes; RRA, robust rank aggregation.

69 downregulated DEGs. A smaller *P*-value in the RRA analysis was reflected by a higher gene rank of differential gene expression. The heatmap shown in Fig. 2B displays the top 10 upregulated and downregulated genes. Enrichment analyses, including GO and KEGG, were used to explore the biological function of the 164 DEGs. GO analysis revealed that the upregulated DEGs were mainly enriched in neutrophil migration and chemotaxis, positive regulation of NF-κB TF activity, and *RAGE and Toll-like receptor binding*. The downregulated DEGs were primarily enriched in Lymphocyte and T cell differentiation as well as T cell receptor complex and binding (Fig. 2C). KEGG pathway enrichment analysis showed that the predominant upregulated terms were IL-17, Toll-like receptor, and TNF signaling pathways, while the primary downregulated terms were T cell differentiation and T cell receptor signaling pathway (Fig. 2D). Collectively, these results suggest that the activation of pro-inflammatory signaling pathways may be concurrent with the inhibition of T cell differentiation and associated pathways in patients with sepsis.

## Gene set enrichment analysis of statistically changed genes based on RRA

To overcome the limitations of differential enrichment analyses, such as the possibility of missing many moderate but meaningful changes due to an inappropriate threshold being chosen (*Geng et al., 2022*; *Xia & Wishart, 2010*), gene set enrichment analysis (GSEA) was conducted using c2.cp.reactome.v2022.1.Hs.symbols.gmt as the reference gene list. RRA analysis screened 2,763 genes with an adjusted $P < 0.05$. Ridgeline and GSEA plots were used to display the distribution of gene expression in the top six most significant gene sets. Notably, it was observed that the activated pathways were predominantly enriched in neutrophil degranulation (NES = 3.104, $P = 1 \times 10^{-10}$, $q = 2.005 \times 10^{-8}$) and innate immune system (NES = 2.604, $P = 1 \times 10^{-10}$, $q = 2.005 \times 10^{-8}$), while the inhibited pathway was enriched in co-stimulation mediated by the CD28 family (NES = −2.734, $P = 2.002 \times 10^{-6}$, $q = 2.027 \times 10^{-4}$; Figs. 3A and 3B). These findings indicated that innate immune responses may be activated and adaptive immunity may be inhibited in sepsis.

## Immune landscape of peripheral blood in patients with sepsis

Among the above enrichment results, significantly changed genes were involved in several immune-related biological functions. Therefore, immune cell infiltration was evaluated across the aforementioned six datasets *via* CIBERSORT analysis. The distribution of 22 immune cell types is illustrated by stacked bar charts in Figs. 4A–4F. Cell types that were not present in all samples were filtered out, and their relative proportions were presented using boxplots (Figs. 4A–4F). Notably, the findings showed that the peripheral blood in patients with sepsis was infiltrated by higher proportions of monocytes, M0 macrophages, and neutrophils in the majority of datasets (≥4 datasets), but lower proportions of CD8 T cells, CD4 T cells, plasma, and resting natural killer cells. These results suggest that sepsis induces a reshaping of the composition and distribution of immune cells in the peripheral blood.

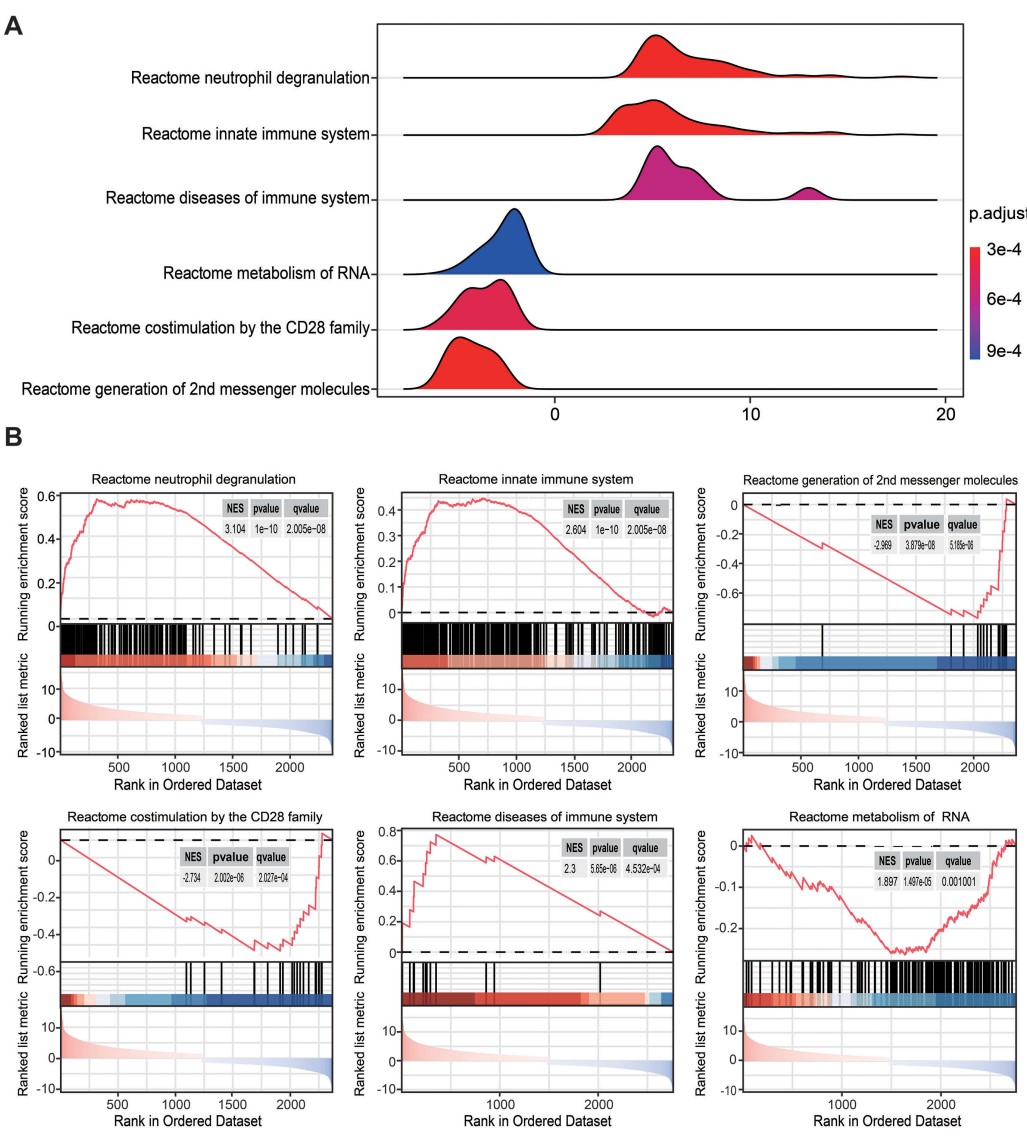

**Figure 3** **Gene set enrichment analysis annotation of statistically changed genes in robust rank aggregation analysis.** (A) Ridgeline plot displaying the distribution of gene expression in the top six most significant gene sets. (B) Gene sets enriched in neutrophil degranulation (NES = 3.104, $P = 1 \times 10^{-10}$, $q = 2.005 \times 10^{-8}$), innate immune system (NES = 2.604, $P = 1 \times 10^{-10}$, $q = 2.005 \times 10^{-8}$), second messenger molecules (NES = $-2.969$, $P = 3.879 \times 10^{-8}$, $q = 3.185 \times 10^{-6}$), co-stimulation by CD28 family (NES = $-2.734$, $P = 2.002 \times 10^{-6}$, $q = 2.027 \times 10^{-4}$), diseases of the immune system (NES = 2.300, $P = 5.65 \times 10^{-6}$, $q = 4.253 \times 10^{-4}$), metabolism of RNA (NES = 1.897, $P = 1.479 \times 10^{-5}$, $q = 0.001$). NES, normalized enrichment score.

## The advanced identification of the key gene set related to sepsis through WGCNA

WGCNA was used to analyze the GSE69528 microarray dataset, which comprises the largest number of samples within the six datasets, namely 83 patients with sepsis and 28 healthy controls. The present study constructed an adjacency matrix that followed a

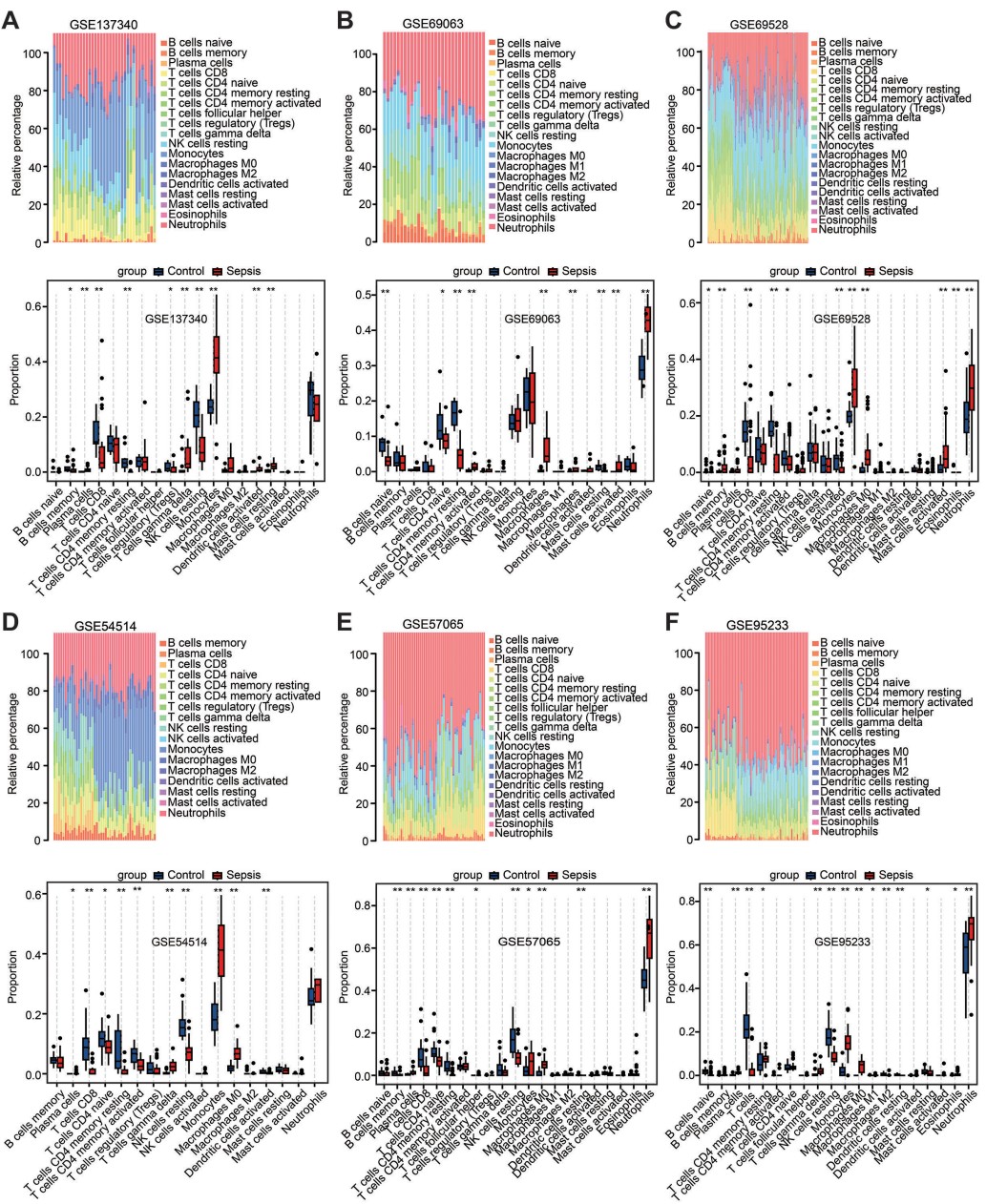

**Figure 4  Estimation of the infiltration of immune cells in the six training datasets using the CIBERSORT analysis.** (A–F) Stacked bar charts revealed the percentage of immune cells in six datasets (A) GSE137340, (B) GSE69063, (C) GSE69528, (D) GSE54514, (E) GSE57065 and (F) GSE95233, and boxplots revealed the differential infiltration of immune cells between patients with sepsis and healthy controls in six datasets (adjusted *P*-values). *$P < 0.05$, **$P < 0.01$, ***$P < 0.001$.

scale-free network by setting $\beta = 7$ ($R^2 = 0.85$) and maintaining high connectivity based on gene distribution (Fig. 5A). The hierarchical clustering method was used to construct gene clusters, and 11 co-expression modules were screened (Fig. 5B). Heatmap visualization revealed that the blue module exhibited the most significant positive correlation with sepsis,

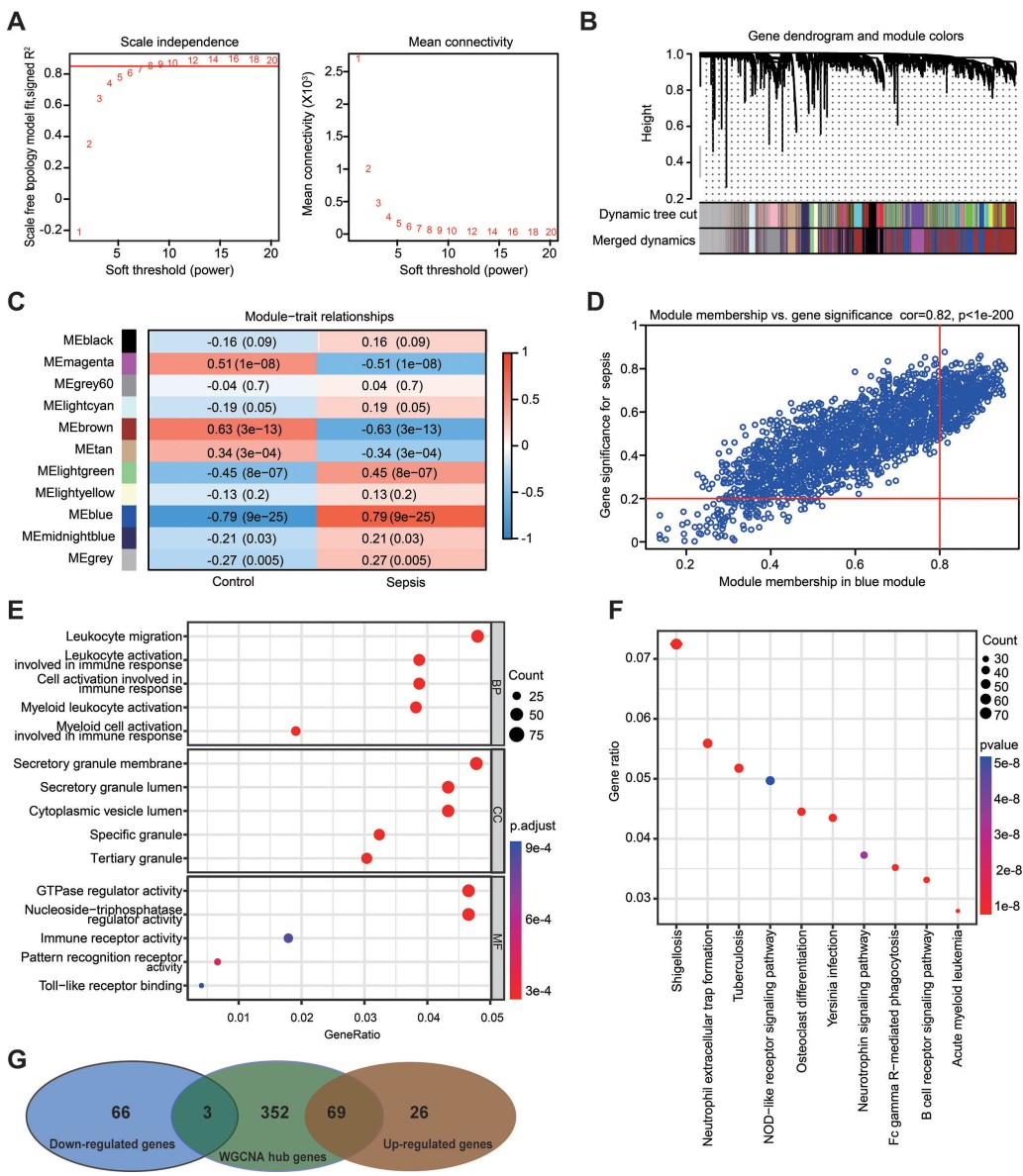

**Figure 5** **Weighted gene co-expression network analysis in the GSE69528 dataset.** (A) Establishment of soft-thresholding power and its mean connectivity. (B) Hierarchical clustering tree showing 11 cluster modules. (C) Heatmap showing the correlation of each module with sepsis. Rows represent modules while columns represent clinical phenotypes. Red indicates a positive correlation, while blue indicates a negative correlation. (D) Scatterplot showing the association between gene significance and module membership in the blue module. (E) Key genes in the blue module were subjected to gene ontology enrichment analysis. (F) Key genes in the blue module were subjected to kyoto encyclopedia of genes and genomes enrichment analysis. (G) Venn diagram showing the obtained gene set.

whereas the brown module demonstrated the strongest negative correlation with sepsis (Fig. 5C). Correlation analysis between gene significance (GS) and module membership (MM) revealed a strong association between the selected module genes and the blue MM ($R = 0.82$, $P < 1 \times 10^{-200}$; Fig. 5D), implying that genes within the blue module exhibited

a strong association with sepsis. Based on the standard of MM > 0.8 and GS > 0.2, 424 genes were selected from the blue module, and their biological functions were revealed using GO and KEGG analyses (Figs. 5E and 5F). The results demonstrated significant enrichment in immune-related terms. The most significantly enriched GO terms for biological processes included leukocyte migration, activation of leukocytes, cells, and myeloid cells involved in immune response activation. For molecular functions, immune receptor activity and pattern recognition receptor activity were significantly enriched. Additionally, the primarily enriched KEGG pathways were neutrophil extracellular trap formation and NOD-like receptor signaling pathway. Ultimately, a gene set including 72 DEGs was obtained by overlapping 164 DEGs from the RRA analysis and 424 genes selected in the WGCNA (Fig. 5G).

## The screening of hub genes in the key gene set *via* PPI analysis combined with Cytoscape software

Based on proteins encoded by the 72 DEGs in the gene set, a PPI network was constructed using the STRING online website (https://cn.string-db.org/). Subsequently, Cytoscape software was used to visualize the interaction network, which consisted of 43 nodes and 62 edges (Fig. 6A). The gene situated in the key node was regarded as a pivotal gene that potentially played a vital physiological function. Important gene modules were identified *via* the plug-in MCODE. By using default parameters, two gene clusters were screened—Cluster 1 and Cluster 2—and the genes in the two modules were labeled as Set 1 (Fig. 6B). Additionally, the plug-in cytoHubba-MCC was used to identify the top 10 genes (which were labeled as Set 2) and their scores (Figs. 6C and 6D).

It is widely acknowledged that gene expression is regulated in a spatiotemporal manner and forms intricate networks involving the interaction between transcription factors (TFs) and their direct target genes (*Xu et al., 2022*). Regulatory processes have important implications in various biological processes such as development, homeostasis, and pathogenesis (*Xu et al., 2022*). The present study employed the plug-in iRegulon of Cytoscape to investigate TFs that regulate Set 2. The findings revealed that TF ELF1 was responsible for regulating the eight genes within Set 2, excluding HP and MCEMP1 (Fig. 6E). By intersecting these eight genes with Set 1, five hub genes were screened: GPR84, hexokinase 3 (HK3), S100A12, CLEC4D and CLEC5A. The expression of these five hub genes was validated by conducting differential expression analysis in the validation dataset GSE28750 and by performing RT-qPCR analysis using blood samples from individuals with sepsis and healthy controls (Figs. 6F and 6G). The results were consistent with the prediction. Additionally, the expression of the predicted TF ELF1 was confirmed through RT-qPCR analysis in peripheral blood monocytes from patients with sepsis and mice after CLP (Table S2A and Fig. 2B).

## The prognostic value evaluation of the hub genes by ROC analysis

If the DEGs identified in sepsis were of clinical significance, they should be indicative of disease severity and thus capable of predicting clinical outcomes. To evaluate the prognostic significance of the five hub genes, ROC analysis was performed in GSE54514,

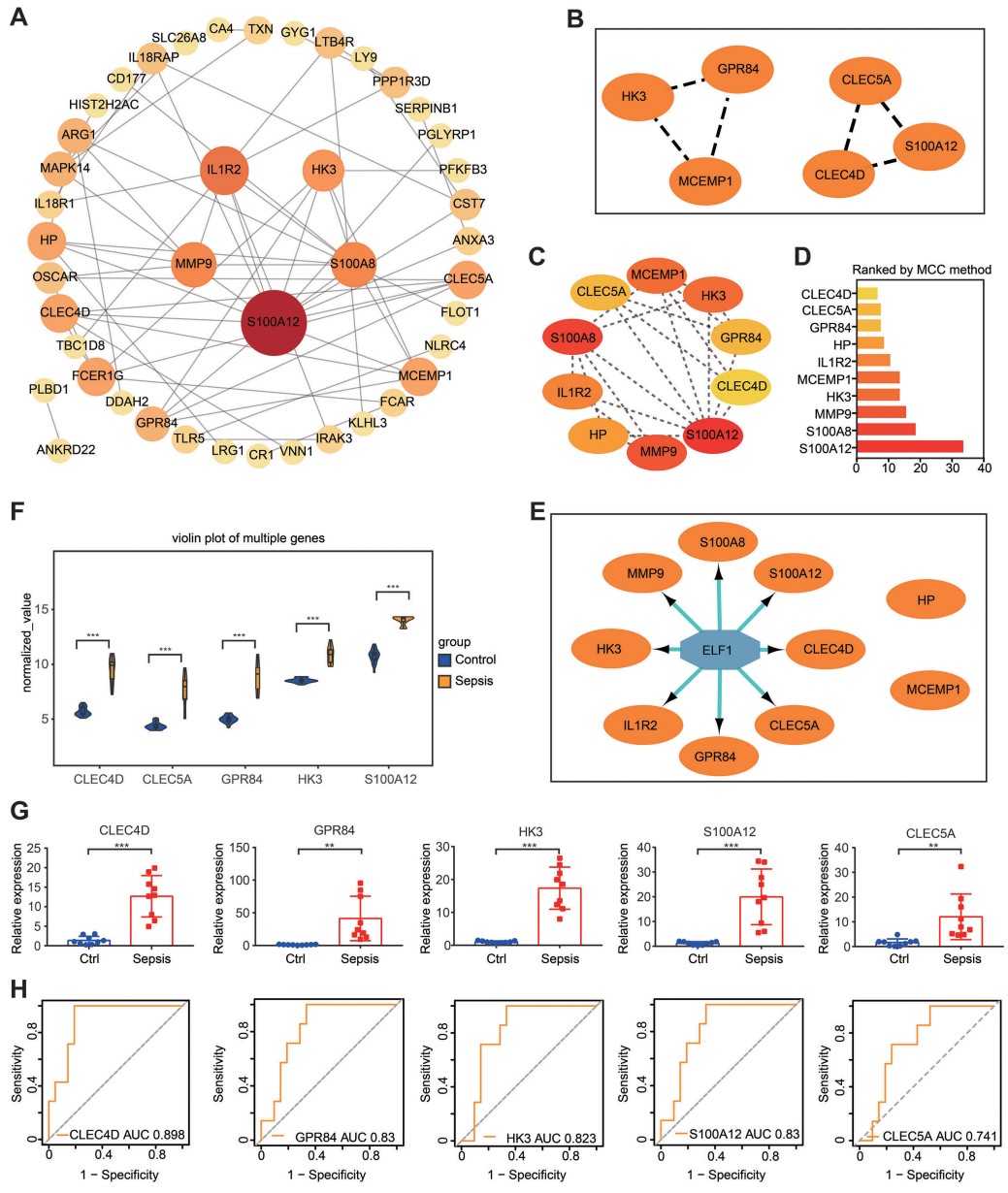

**Figure 6  PPI network construction and hub gene identification.** (A) A PPI network was constructed and visualized using Cytoscape. The nodes represent proteins, while the edges represent their interactions. (B) The plug-in Minimal Common Oncology Data Elements screened two subnetworks: Cluster 1 and Cluster 2. (C and D) The plug-in cytoHubba- Maximal Clique Centrality was performed to identify the top 10 genes in the network. (E) The plug-in iRegulon highlights the predicted transcription factor in blue, while target genes are shown in orange. (F) Analysis of differential gene expression in the GSE28750 validation dataset. ***$P < 0.001$. (G) The expression of hub genes was validated in peripheral blood samples from patients with sepsis and healthy controls using reverse transcription-quantitative PCR. **$P < 0.01$, ***$P < 0.001$. (H) Receiver operating characteristic curve showing prognosis. PPI, protein–protein interaction.

which includes survivor and non-survivor data. Unexpectedly, the prognosis of sepsis was significantly associated with four of the five hub genes: HK3, GPR84, CLEC4D, and S100A12. Among these genes, CLEC4D exhibited the highest prognostic value (AUC = 0.898) in distinguishing between sepsis survivors and non-survivors. The prognostic values of the other genes were as follows: GPR84 (AUC = 0.830), S100A12 (AUC = 0.830), and HK3 (AUC = 0.823), and CLEC5A (AUC = 0.741, a moderate prognostic value; Fig. 6H). These four genes were designated as the hub genes with prognostic significance.

## The correlation analysis of the hub genes in patients with sepsis

Correlation analysis of the expression data from the GSE28750 validation cohort was conducted to investigate the expression patterns of the four significant hub genes. Prior to the analysis, batch effect removal, normalization, and log2 transformation were applied to process the dataset (Fig. S3A and Fig. 3B). A robust positive correlation was observed among the four hub genes (Fig. 7A). This result indicated that the four significant hub genes were potentially regulated by a shared TF. Additionally, the associations between the hub genes and immune cells were explored. The hub genes exhibited a strong positive association with innate immune cells including neutrophils, activated DCs, and M0 macrophages. Conversely, the hub genes demonstrated a negative association with acquired immune cells including B, CD4 T cells, and CD8 T cells (Fig. 7B). Moreover, most immune genes showed a negative correlation with the hub genes (Fig. 7C). The findings indicated that the four hub genes may be closely associated with immune dysfunction in septic patients.

## Construction of a ceRNA network regulating the expression of hub genes

Extensive research on miRNAs is driven by their potential as therapeutic targets for a wide range of diseases. miRNAs can downregulate gene expression and induce gene silencing by binding mRNA molecules. However, lncRNAs, acting as upstream molecules, can influence miRNA function by interacting with miRNA response elements (MREs), leading to the upregulation of gene expression. This interaction between RNAs is called ceRNA (Li et al., 2018). Four online miRNA databases with default parameters were used to identify miRNAs that regulate ELF1 activity. Subsequently, miRNAs that were found to overlap in three databases were selected. In total, 98 miRNAs targeting ELF1 were screened, and a network was constructed using Cytoscape software (Fig. 8A).

The StarBase 2.0 database was used to investigate the interactions between the 98 miRNAs and lncRNAs, and a comprehensive literature search was also conducted. Based on the ceRNA hypothesis, one downregulated miRNA and one upregulated lncRNA previously reported in sepsis were selected (Meng et al., 2023), resulting in the proposal of the following ceRNA network: NEAT1-*Homo sapiens* (hsa)-miR-495-3p-ELF1. This network was visualized using Cytoscape software and is presented in Fig. 8B. It is speculated that this ceRNA network may serve as a critical regulatory pathway in sepsis.

## DISCUSSION

As the pathogenesis of sepsis has yet to be fully elucidated, it is necessary to further clarify the intricate molecular mechanisms underlying this condition. In recent years, the combination

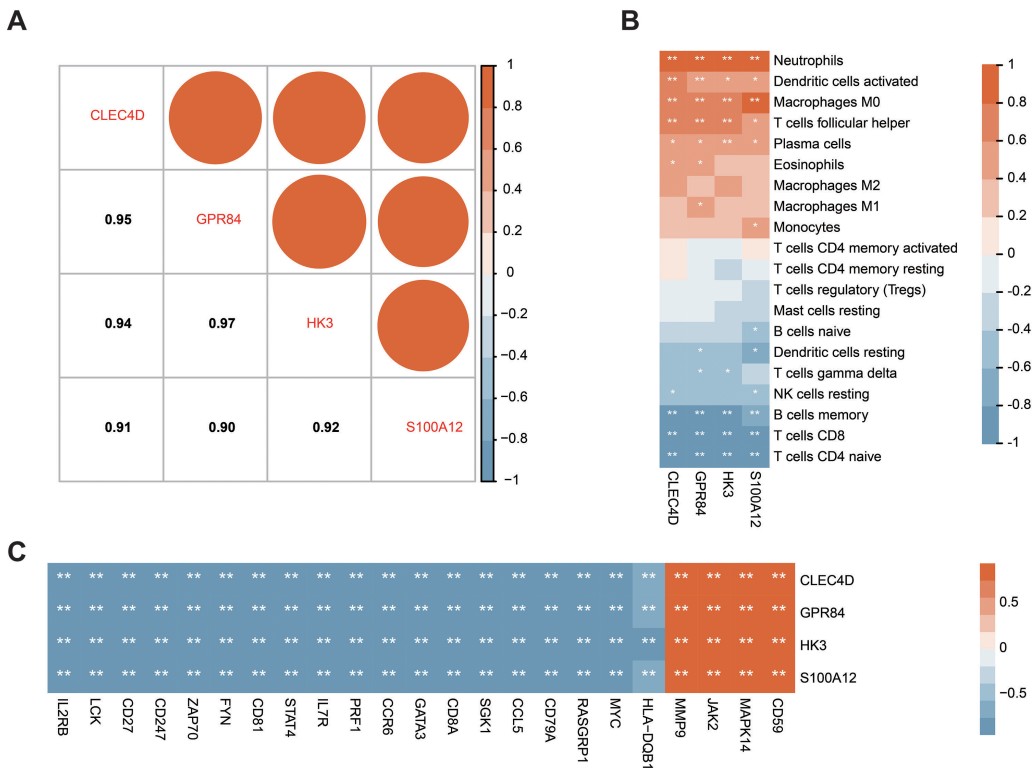

**Figure 7 Correlation between the prognostic hub genes and immune cell types as well as immune genes.** (A) Correlation analysis of the four prognostic hub genes. (B) Analysis of the correlation between the four prognostic hub genes and immune cells. (C) Analysis of the correlation between the four prognostic hub genes and immune genes. *$P < 0.05$, **$P < 0.01$.

of high-throughput sequencing technology and bioinformatics analysis has become a widely adopted strategy for exploring various diseases. This strategy not only sheds light on the molecular mechanisms underlying disease pathogenesis but also facilitates the discovery of potential therapeutic targets (*Gong et al., 2022*). Numerous bioinformatics methods have been employed to identify potential key molecules or regulatory mechanisms in sepsis over the past few decades (*Gong et al., 2022*; *Zeng et al., 2021*); however, the RRA method has not been used in previous research. The RRA algorithm possesses many advantages, such as the capability to process data from diverse sources, robustness against noise interference, and the ability to handle incomplete rankings (*Chen et al., 2022*; *Jia & Zhai, 2019*). The present study integrated datasets from diverse chip platforms using RRA and screened transcriptome data of peripheral blood in 232 septic patients and 116 healthy volunteers. This contributed to augmenting the sample size and mitigating potential biases in the interpretation of results.

The RRA analysis identified 1,487 upregulated genes and 1,276 downregulated genes with statistical significance in sepsis. GSEA analysis revealed that major signaling pathways associated with the innate immune system such as neutrophil degranulation were activated, but that signaling pathways related to the adaptive immune system such as co-stimulation

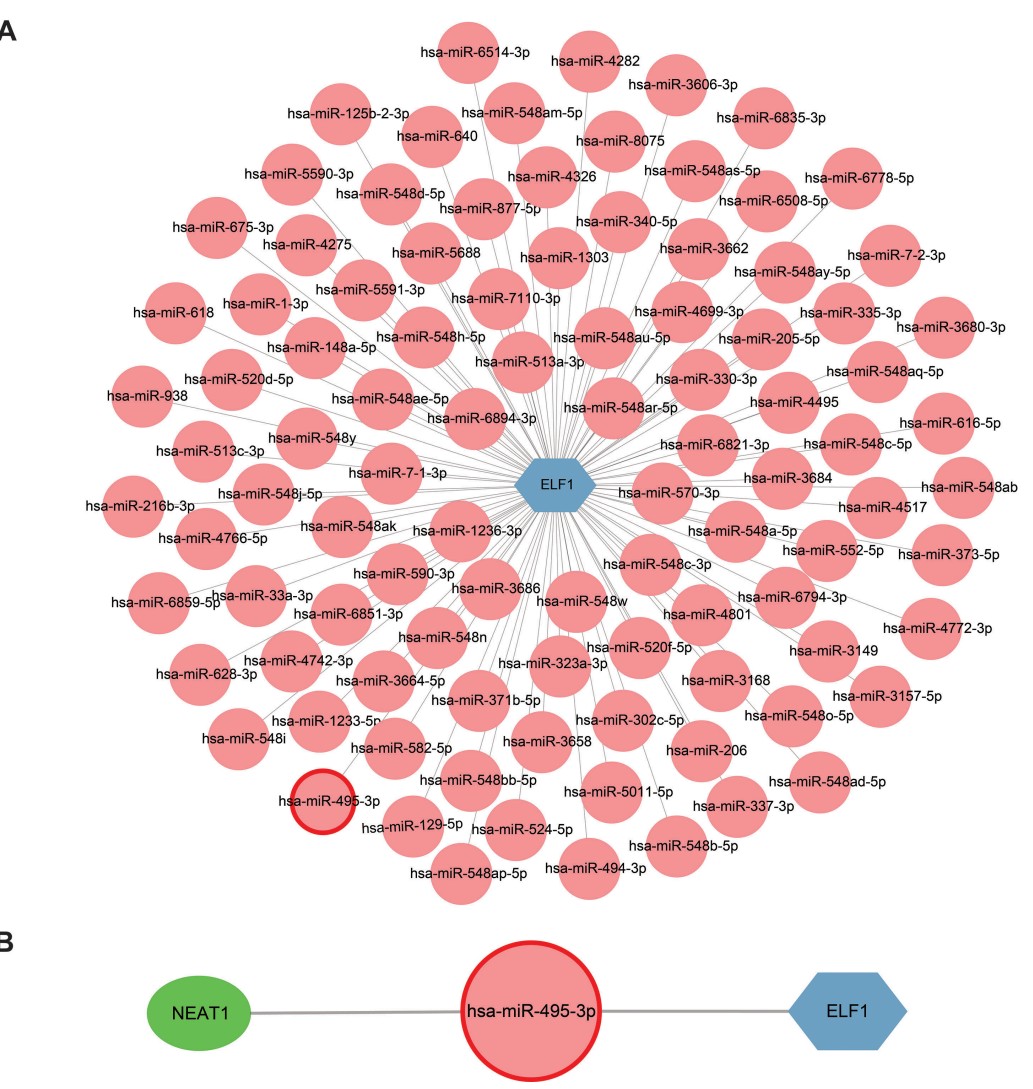

**Figure 8 Construction of a miRNA-mRNA network and a ceRNA network.** (A) ELF1 (blue hexagon) and 98 regulatory miRNAs (pink circle) formed the miRNA-mRNA network. (B) Construction of a ceRNA network where the green elliptical, pink round node and blue hexagonal nodes represent long noncoding RNA, miRNA and mRNA, respectively . miRNA, microRNA; ceRNA, competitive endogenous RNA.

by the CD28 family and generation of second messenger molecules were significantly inhibited (Table S2). Stricter standards were used to screen the statistically changed genes through six training cohorts, and this identified 95 upregulated and 69 downregulated DEGs in septic patients compared to healthy controls. Consistent with the above results, functional enrichment analyses revealed that the most enriched terms by downregulated DEGs were involved in adaptive immunity such as T cell receptor binding and signaling pathway and T helper (Th)1, Th2 and Th17 cell differentiation (Figs. 2C and 2D). However, upregulated DEGs were primarily involved in *neutrophil migration and chemotaxis, IL-17*

*signaling pathway, and TNF signaling pathway* (Figs. 2C and 2D), which are all associated with pro-inflammatory responses and contribute to a hyper-inflammatory state.

CIBERSORT analysis showed an increase in the percentage of innate immune cells, including monocytes, M0 macrophages, and neutrophils, and a decrease in the percentage of acquired immune cells such as plasma cells, CD8 T cells, and CD4 T cells (Figs. 4A–4F). The correlation analysis also demonstrated that the hub genes with prognosis value had a robust positive correlation with innate immune cells and a significant negative correlation with adaptive immune cells (Fig. 7B). The sepsis characterized by leukocytosis shows a significant increase in neutrophils and monocytes, followed by the state of lymphopenia including B cells, CD4 T cells, and CD8 T cells (*Chen et al., 2022*; *Van der Poll et al., 2017*). Additionally, *Chen et al. (2022)* demonstrated that septic patients could develop immune suppression due to T cell exhaustion and dysfunctional T cell repertoire. Sepsis data from the current study demonstrated that the state of hyperinflammation was associated with the increase of innate immune cells and activation of their pathways, but that immunosuppression was related to the decrease of adaptive immune cells and inhibition of their pathways. It is universally accepted that the hyperinflammation state and the immune suppression state are completely distinct (*Zhang et al., 2023*). However, a recent study has found that the host response in patients with sepsis showed signs of concurrent hyperinflammation and immune suppression involving different cell types and organ systems once the patients were hospitalized (*Van der Poll et al., 2017*), but the mechanisms of this condition are unknown. The current study's findings lay a theoretical foundation for further understanding this phenomenon.

Based on a comprehensive analysis involving RRA, WGCNA, and Cytoscape, five hub genes (CLEC4D, GPR84, S100A12, HK3, and CLEC5A) were screened, and their expression was validated in the GSE28750 validation cohort and in peripheral blood samples. Subsequently, ROC analysis identified four hub genes (CLEC4D, GPR84, S100A12, and HK3) with prognostic value. Notably, CLEC4D had the highest prognostic value in distinguishing between sepsis survivors and non-survivors (AUC = 0.898). CLEC4D belongs to the C-type lectin (CTL)/CTL domain superfamily (*Miyake & Yamasaki, 2020*). Its associated pathways include the innate immune system and CLEC7A (also known as Dectin-1) signaling. CLEC4D also functions as an endocytic receptor and serves as a pattern-recognition receptor of the innate immune system, capable of identifying damage-associated molecular patterns (DAMPs) and pathogen-associated molecular patterns derived from bacteria and fungi. This recognition triggers the activation of NF-$\kappa$B, the maturation of antigen-presenting cells, and the transition of T cells (*Drouin, Saenz & Chiffoleau, 2020*; *Miyake, Masatsugu & Yamasaki, 2015*; *Miyake & Yamasaki, 2020*). Of note, a previous study demonstrated differential expression of CLEC4D between pediatric patients with sepsis and controls (*Bai et al., 2020*), similar to the present findings.

Regarding the other hub genes, GPR84, an orphan G protein-coupled receptor, does not have a known physiological ligand (*Luscombe et al., 2020*). Its expression has been primarily observed in hematopoietic tissues, particularly bone marrow (*Luscombe et al., 2020*). Previous studies identified GPR84 as a receptor for medium-chain fatty acids with carbon chain lengths ranging from nine to 14 that could be induced in monocytes/macrophages

upon activation by lipopolysaccharide. However, the physiological function of GPR84 has not been fully elucidated (*Forsman et al., 2023*; *Luscombe et al., 2020*). S100 calcium-binding protein A12 (S100A12) is a calcium-, zinc-, and copper-binding protein that plays a prominent role in the regulation of inflammatory processes and the immune response. Its pro-inflammatory activity involves the recruitment of leukocytes, the promotion of cytokine and chemokine production, and the regulation of leukocyte adhesion and migration. Additionally, it acts as an alarmin or a DAMP molecule, and stimulates innate immune cells by binding to receptors for advanced glycation end products (*Carvalho et al., 2020*; *Xia et al., 2023*). According to a previous study, high plasma values of S100A12 at hospital admission can indicate a higher risk of mortality in patients with septic shock (*Dubois et al., 2019*). HK3 is an enzyme that catalyzes the phosphorylation of hexose to hexose 6-phosphate and mediates the initial step of glycolysis by catalyzing the phosphorylation of D-glucose to D-glucose 6-phosphate (*Seiler et al., 2022*). A recent study also indicated that HK3 was a diagnostic marker and had potential functions in pediatric sepsis (*Zhang et al., 2021*). This result is similar to the present findings. Therefore, by participating in various pathways such as antigen recognition and immune cell metabolism, the four hub genes with prognostic value identified in the current study (CLEC4D, GPR84, S100A12, and HK3) appear to play a crucial role in the occurrence and development of inflammation, ultimately impacting patient prognosis.

miRNAs are a class of endogenous, short non-coding RNAs that are extensively studied for their capability to modulate the stability and translation of mRNAs by binding the 3′-untranslated region of mRNAs. The construction of ceRNA networks has gained great interest in recent years. Such networks use miRNAs as bridges to establish the association between downstream target mRNAs and upstream lncRNAs by combining MREs (*Cheng et al., 2021*). Identification of key miRNAs and lncRNAs involved in sepsis pathogenesis should provide new strategies for disease diagnosis and treatment (*Cheng et al., 2021*). The present study validated that the mRNA expression of ELF1, which was predicted as a TF of the hub genes, was upregulated in patients with sepsis and in a mouse model of CLP (Table S2A and Fig. 2B). Consistent with the present results, *Xia et al. (2023)* revealed that the mRNA expression of ELF1 was upregulated *in vivo* and *in vitro* in sepsis. The present study conducted a comprehensive literature search to identify dysregulated lncRNAs and miRNAs in sepsis based on the aforementioned ceRNA hypothesis. A recent study revealed that the expression of NEAT1 was significantly increased while the expression of miR-495-3p was decreased in burn-associated sepsis and that the expression of these two molecules was associated with the clinical features of patients (*Meng et al., 2023*). Other studies also indicated that the expression of NEAT1 and miR-495-3p changed in sepsis and that NEAT1 exhibited great potential for sepsis diagnosis and treatment, considering its modification of miR-495-3p in inflammatory cell models (*Ghafouri-Fard et al., 2021*; *Xia et al., 2020*). In addition, it was reported that the lncRNA NEAT1 was upregulated in peripheral blood mononuclear cells (*Shui et al., 2019*). By integrating the predictive results from five online miRNA websites, the current study proposed a potential lncRNA-miRNA-mRNA pathway: the NEAT1-hsa-miR-495-3p-ELF1 pathway, which may exert an important influence on the pathogenesis of sepsis.

To summarize, the present study first used an RRA approach to integrate multiple transcriptome datasets derived from peripheral blood in sepsis. Through a series of bioinformatic methods and validation of RT- qPCR of peripheral blood, this study screened four hub genes with high prognostic value and constructed a ceRNA network regulating their expression. Additionally, the study demonstrated the phenomenon of concurrent hyperinflammation and immunosuppression in septic patients. The present study may provide helpful prognostic and therapeutic clues to support the ongoing battle against sepsis.

### Funding

This study was supported by grants from the National Natural Science Foundation of China (nos. 82272194, 82002089, 82130063 and 82241061), Guang Dong Basic and Applied Basic Research Foundation (no. 2022A1515140029). The funders had no role in study design, data collection and analysis, decision to publish, or preparation of the manuscript.

### Grant Disclosures

The following grant information was disclosed by the authors:
National Natural Science Foundation of China: 82272194, 82002089, 82130063, 82241061.
Guang Dong Basic and Applied Basic Research Foundation: 2022A1515140029.

### Competing Interests

The authors declare there are no competing interests.

### Author Contributions

- Kangyi Xue conceived and designed the experiments, performed the experiments, analyzed the data, prepared figures and/or tables, authored or reviewed drafts of the article, and approved the final draft.
- Kan Wu performed the experiments, prepared figures and/or tables, and approved the final draft.
- Haoxian Luo performed the experiments, prepared figures and/or tables, and approved the final draft.
- Haihua Luo performed the experiments, analyzed the data, authored or reviewed drafts of the article, and approved the final draft.
- Zhaoqian Zhong performed the experiments, analyzed the data, prepared figures and/or tables, and approved the final draft.
- Fen Li conceived and designed the experiments, authored or reviewed drafts of the article, and approved the final draft.
- Lei Li conceived and designed the experiments, analyzed the data, authored or reviewed drafts of the article, and approved the final draft.
- Li Chen conceived and designed the experiments, analyzed the data, prepared figures and/or tables, authored or reviewed drafts of the article, and approved the final draft.

## Human Ethics

The following information was supplied relating to ethical approvals (*i.e.*, approving body and any reference numbers): Clinical Trial Ethics Committee, The Third Affiliated Hospital of Southern Medical University (2023073).

## Animal Ethics

The following information was supplied relating to ethical approvals (*i.e.,* approving body and any reference numbers): The Institutional Animal Care and Use Committee of Southern Medical University (approval no. L2018235).

## Data Availability

The datasets are available at GEO: GSE137340, GSE69063, GSE69528, GSE54514, GSE57065, GSE95233, and GSE28570.

## Supplemental Information

Supplemental information for this article can be found online at http://dx.doi.org/10.7717/peerj.19619#supplemental-information.

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
