# Peer review of "Identification of hub genes and prediction of the ceRNA network in adult sepsis"

_PeerJ, doi:10.7717/peerj.19619_

## Round 0.1 · original submission · Major Revisions

Please address the concerns of all reviewers and revise the manuscript accordingly.

**Language Note:** The review process has identified that the English language must be improved. PeerJ can provide language editing services - please contact us at [email protected] for pricing (be sure to provide your manuscript number and title). Alternatively, you should make your own arrangements to improve the language quality and provide details in your response letter. – PeerJ Staff

·

Basic reporting

Thank you for inviting me to review the submitted manuscript titled " Identification of hub genes and prediction of the ceRNA network in adult sepsis". Briefly, this is an interesting and meaning study based on the bioinformatics analyses. The authors used several GEO databases, and made kinds of bioinformatics analyses. The authors also validated the key hub genes using qPCR. Overall, this is a good study.

Experimental design

no comment

Validity of the findings

no comment

Additional comments

My only minor concerns is the figure quality. Several figures need to be improved more carefully. For example, in figure 5B, the letter "t" and "r" are getting together. Additionally, the key bioinformatics messages from GO and KEGG analyses should be highlighted.

Reviewer 2 ·

Basic reporting

Thanks for the opportunity to review your study. I have several comments
1. The present study seems to be very general. The authors have used several bioinformatics tools to explore the known information. In the present study, integrated bioinformatics analyses were conducted, and validation dependent on patient datasets and blood samples was performed. The major drawback of the study is the lack of supportive data (patient samples data) to prove the concept of the study.
2. The study goal is very general: to test your measurements against several outcomes through a bioinformatics tool. Your abstract states that hyperinflammation was closely associated with the innate immune system, while immunosuppression primarily involved the adaptive immune system. It is not true. Patients with sepsis and septic shock have hyperinflammatory responses associated with both arms of immunity. The majority of sepsis patients died due to multiple organ failure has severe defects in the adaptive arms of immunity.
3. The study highlighted five hub genes in the module that were identified and verified by the above six training cohorts and RT- qPCR of peripheral blood in septic patients. The comparative data of the folds change expression in patients with sepsis and healthy controls was done in just nine samples each. It is not a sufficient sample size to conclude the findings of the study.
4. Did the author calculate the power of study?
5. Because of the general study goal, the discussion remains very descriptive explaining that the markers were somewhat different between the groups. However, it would be more interesting whether the biomarkers are as distinct as in the in-vitro experiment or patient samples and how you answer your study questions.

Experimental design

It is purely a general study design marginally lacking the experimental works

Validity of the findings

The concept of the study is proved by advanced bioinformatics tools.

Additional comments

The study has several grammatical mistakes and typos.

Reviewer 3 ·

Basic reporting

It is important study , however the aim was not adequately clarified ., the importance of the sepsis problem is well illustrated .
The authors need to write clear aim /objective as well as research questions and hypothesis.
In the method section , the inclusion criteria have to be more clarified , .explain why need for 10 samples and how the sepsis will be more than 5 sepsis . and also the control , as this will end in more than 10 .
The technical part of the study is well written .
Some figures as number4 , is not clear

Experimental design

The methods is well illustrated .

Validity of the findings

Authors need to elaborate on the reproducibility of the findings .

Additional comments

Non

---

## Round 0.2 · accepted · Accept

All issues raised by the reviewers were addressed and the revised manuscript is acceptable now.

·

Basic reporting

Thanks for addressing my concerns.

Experimental design

no comment

Validity of the findings

no comment

Additional comments

no comment

Reviewer 3 ·

Basic reporting

This is the second revision .The authors have responded to my comments with proper changes in the manuscript .

Experimental design

This is the second revision .The authors have responded to my comments with proper changes in the manuscript .

Validity of the findings

This is the second revision .The authors have responded to my comments with proper changes in the manuscript .